🔓 | **Open Peer Review** | Bacteriology | Research Article

# A GntR family regulator HutC senses PCA to regulate histidine catabolism in *Pseudomonas aeruginosa*

Guoyan Cui,[1] Zhuang Li,[2] Yiqiang Zhang,[3] Fengping Wang,[4] Dan Li,[4] Yujiao Zhao,[4] Jia Cui[1]

**ABSTRACT** Pathogenic bacteria, *Pseudomonas aeruginosa*, have evolved various regulatory mechanisms to cope with the adverse environments. HutC is associated with the regulation of histidine metabolism in bacteria. Despite extensive research on histidine metabolism, the identification of the HutC regulon is not systematic or comprehensive. In this study, we demonstrated that the *hut* operon including *hutF-CHUIG* is involved in histidine catabolism in *P. aeruginosa*. We also identified that HutC participates in the regulation of histidine catabolism. Promoter activity measurement uncovered that HutC directly binds to the *hutFHUIG* promoter region and represses its transcription. Electrophoretic mobility shift assay with recombinant HutC showed that HutC directly represses the transcription of *hutIG*. More importantly, this inhibitory effect was relieved by the addition of phenazine-1-carboxylic acid. Collectively, these findings greatly expand our understanding of HutC as an important metabolic sensor for regulating Hut system in order to adapt to the harsh environments in *P. aeruginosa*.

**IMPORTANCE** Environmental signaling molecules are pivotal mediators that enable bacteria to sense fluctuating environmental conditions, coordinate critical physiological processes including growth regulation, metabolic reprogramming, and virulence modulation. Our research elucidates two pivotal breakthroughs in bacterial environmental signal sensing mechanisms: First, we elucidated the regulatory mechanism of HutC in histidine catabolism through genetic and biochemical analysis. Second, utilizing untargeted metabolomics coupled with EMSA validation, we discovered PCA as a novel signaling molecule that interacts with HutC to modulate histidine catabolism. These findings not only expand our understanding of microbial cross-talk with environmental stimuli but also highlight its significance in both bacterial ecological adaptation and pathogenesis. The identification of PCA as a metabolic ligand opens new avenues for developing anti-infective strategies focused on bacterial signaling networks, while providing a methodological framework for uncovering latent environmental sensors in microbial communities.

**KEYWORDS** *Pseudomonas aeruginosa*, HutC, histidine catabolism, phenazine-1-carboxylic acid

**Peer Reviewers** Stephen Dela Ahator, National University of Singapore, Singapore, Singapore; Paul Brown, University of the West Indies, Kingston, Jamaica

Address correspondence to Jia Cui, cuijia@czmc.edu.cn.

Guoyan Cui and Zhuang Li contributed equally to this article. The author order was determined in order of increasing seniority.

The authors declare no conflict of interest.

See the funding table on p. 14.

*P*seudomonas aeruginosa is a ubiquitous opportunistic pathogen with large transcriptional regulatory networks that coordinate cellular processes for effective response to diverse environments. The high adaptability results from large genome sizes and high levels of metabolic network redundancy (1). The synthesis and degradation pathways of histidine, as the fourth most energetically expensive amino acid to synthesize, are tightly controlled in bacteria (2). When histidine is in excess, many bacteria catabolize this amino acid for carbon, nitrogen, and energy generation via the histidine utilization (Hut) system (3). Histidine catabolism is highly conserved among bacteria. Five enzymes are required to convert histidine to glutamate in *Pseudomonas,* including histidine ammonia-lyase

(HutH), urocanase (HutU); imidazolone propionate amidohydrolase (HutI), ormiminoglutamate deiminase (HutF) and formylglutamate amidohydrolase (HutG) (4). However, in *Salmonella*, *Klebsiella*, and *Bacillus*, histidine degradation requires a "four-step" pathway (5–7). The first three steps of histidine catabolism appear to be the same in bacteria. The first step is that the deamination of histidine by HutH converts histidine into urocanate, which is then processed by the enzyme HutU to imidazolone propionate (IP). And the ring of IP is cleaved to yield formimino-glutamate (FIG) by HutI. The breakdown of FIG differs in different organisms. In *Pseudomonas* and *Brucella,* FIG is converted into glutamate and formate via a two-step enzymatic reaction (8), whereas in enteric bacteria (such as *Salmonella* and *Klebsiella*) and gram-positive bacteria, FIG is directly hydrolyzed to glutamate and formamide by a single enzyme formimino-glutamase (HutE) (9). The five-step pathway provides an extra ammonia molecule compared with the four-step pathway. The evolutionary significance of the two pathways remains unclear because the final results of the two ways are the same. Moreover, the Hut system appears to be absent in *Escherichia*, *Edwardsiella*, *Shigella*, and *Proteus* (9). The absence of the *hut* operon in *E. coli* makes it more difficult due to its importance in generating crucial regulatory paradigms.

Histidine catabolism is highly regulated, and futile production or excessive catabolism is unwarranted because synthesis of histidine is expensive to a cell (10). Induction of the Hut system is controlled by HutC, which is a representative member of the GntR family of transcriptional regulators (11). Similar to other transcriptional regulators, HutC plays an important role in response to environmental signals by modulating the expression of related genes (3, 12). For example, the binding activities to the *hut* operators were dissociated by the effector molecule of urocanate in *P. putida* (13). In this study, we investigated how HutC controls the *hut* operon transcription in response to environmental signals in *P. aeruginosa*. Here, we screened the key metabolites through untargeted metabolomics. Meanwhile, we proved that the absorption peak of standard PCA aligns with the culture supernatant of the Δ*hutC* strain by full-wavelength scanning and HPLC. EMSA also proved that the binding activities to *hutIG* and HutC were abolished by the addition of PCA. In addition, we characterized the *hut* gene cluster by gene deletion and complementation experiments, growth, and biochemical assays. Importantly, we also revealed that HutC responds to environmental signal PCA by activating the gene expression of the *hut* operon.

## MATERIALS AND METHODS

### Bacterial strains and culture conditions

Bacterial strains and plasmids used in this study are detailed in Table S3. *P. aeruginosa* PAO1 and its derivatives were grown at 37°C in Luria-Bertani (LB), or chemically defined minimal medium (M9) supplemented with various carbon and nitrogen sources (10 mM histidine, urocanate, PCA, and 1-HP). For selection and growth, antibiotics were applied at these concentrations: 100 µg/mL ampicillin, 50 µg/mL kanamycin, and 10 µg/mL tetracycline for *E. coli*; 300 µg/mL trimethoprim, 100 µg/mL tetracycline, and 300 µg/mL carbenicillin for *P. aeruginosa*.

### Construction of plasmids

The *hutC* complementation plasmid was created by amplifying DNA fragments containing the *hutC* promoter using p-*hutC*-com-F/R primer pairs (Table S4), which were then inserted into the pAK1900 vector. Parallel cloning strategies generated the p-*hutH/I/G/U/F* series, employing gene-specific primer sets detailed in Table S2 for each respective operon.

To construct the *hutH-lux* fusion plasmid, we used the pMS402 plasmid as the backbone for creating a promoter-*luxCDABE* reporter fusion specific to the *hutH* gene as previously described (14). The promoter region of *hutH* was amplified via PCR with the

primers *hutH-lux*-F/R (Table S4) and cloned into the PMS402. An analogous approach was used to generate the *hutG-lux, hutU-lux,* and *hutF-lux* plasmids.

Plasmid pET-*hutC* was generated by amplifying the target sequence using the primers pET-*hutC*-F/R, thereafter inserted into the pET28a vector. The recombinant plasmid was transferred into *E. coli* BL21 (DE3) competent cells to facilitate the expression of protein.

## Construction of *P. aeruginosa* mutants

For construction of *P. aeruginosa hutC* knockout mutant, a *sacB*-based tactic was adopted as described previously (15). The upstream primer pairs of pEX-*hutC*-up-F/R and the downstream primer pairs of pEX-*hutC*-down-F/R were designed to amplify the homologous sequences (Table S4). The two PCR products were digested with indicated restriction enzymes and fused into pEX18Amp knockout plasmid. Then, the plasmids were electroporated into PAO1 with selection based on carbenicillin resistance. Bacterial isolates underwent dual phenotypic assessment via replica plating: antibiotic resistance profiling against carbenicillin (300 μg/mL) and osmoregulatory capacity evaluation on high-sucrose media (15% wt/vol), both cultivated on LB agar substrates. Finally, the *hutC* knockout mutant after double crossover event was confirmed through PCR using the primer pairs *hutC*-test-F/R. A similar strategy was used to construct the Δ*hutH*, Δ*hutI*, Δ*hutG*, Δ*hutU*, and Δ*hutF*.

## Transcriptome experiments

The wild-type (WT) PAO1 and Δ*hutC* strain were cultured to $OD_{600}$ = 0.6 in LB medium at 37℃. Total RNA was extracted from cell cultures using the CTAB method, and then genomic DNA was removed. Only high-quality RNA samples were selected for library preparation. Ribosomal RNA (rRNA) depletion was achieved using the RiboCop rRNA Depletion Kit for mixed bacterial samples (Lexogen, USA), after which the remaining mRNAs were converted into cDNA library with random hexamer primers (Illumina). RNA-seq transcriptome libraries were prepared following Illumina Stranded mRNA Prep, Ligation (San Diego, CA) using total RNA. Each sample was processed two times. RNA-seq reads were mapped to the *P. aeruginosa* genomes (NC_002516.2) using Bowtie2, and only the uniquely mapped reads were retained for the subsequent transcriptome analyses. All of the analyses were performed on the online platform of Majorbio Cloud Platform (https://www.majorbio.com/). DEGs were identified using DESeq2 (BH-adjusted $P < 0.05$ and log2 fold change > 1). The data have been uploaded to BioProject and accession numbers are PRJNA1199230.

## Quantitative real-time PCR (qRT-PCR)

The WT PAO1, Δ*hutC* mutant, and the complemented strains were cultivated in LB medium at 37℃ until they reached the mid-exponential phase. For RNA extraction, 1 mL each strain's culture was centrifuged at 8,000 × *g* for 5 min. The quality and concentration of the total RNA samples were evaluated by agarose gel electrophoresis and spectrophotometry, respectively. Total RNA was reverse-transcribed to cDNA using reverse transcriptase (TAKARA). qRT-PCR was performed using SYBR Green PCR Supermix (Mei5bio, China) on a Bio-Rad CFX96 Real-Time thermocycler. The relative mRNA levels were normalized by using rpsL as an internal control, and the relative expression was determined using the $\Delta\Delta C_t$ value as previously described (16). Relevant primers used in this study are listed in Table S4.

## Metabolomics and data analysis

*P. aeruginosa* WT PAO1 and Δ*hutC* mutant were grown to late-log phase ($OD_{600} \approx 2.0$) in LB medium at 37℃. For metabolite isolation, 50 mg of cell pellets were transferred to cryovials containing 6 mm grinding beads. A 400 μL methanol:water mixture (4:1 vol/vol) supplemented with 0.02 mg/mL L-2-chlorophenylalanine (internal standard) was added for biphasic extraction. Samples were processed through cryogenic homogenization

(−10°C, 50 Hz, 6 min) using a Wonbio-96c cryomill, followed by ultrasonication (40 kHz, 30 min, 5°C). After phase separation at −20°C (30 min), supernatants were collected post-centrifugation (13,000 × $g$, 15 min, 4°C) for subsequent analysis.

Metabolomics analysis was conducted on a Thermo UHPLC-Q Exactive HF-X system equipped with an ACQUITY HSS T3 column (100 × 2.1 mm, 1.8 μm). The binary solvent system comprised: mobile phase A: 0.1% aqueous formic acid with 5% acetonitrile; mobile phase B: 0.1% formic acid in acetonitrile-isopropanol (1:1 vol/vol). Operational parameters included: 400 μL/min flow rate, 40°C column temperature, and 3 μL injection volume. MS detection featured dual-polarity mode (±3,500 V ISVF) with auxiliary gas heating (425°C) and capillary temperature (325°C). Collision energy gradients (20–60 eV) were applied in data-dependent acquisition mode, maintaining resolutions of 60,000 (full MS) and 7,500 (MS/MS).

Raw UHPLC-MS data were processed through Progenesis QI (Waters, USA) for baseline correction, peak alignment, and feature extraction. Metabolite identification involved cross-referencing against HMDB, Metlin, and the self-built Majorbio Database via the MAJORBIOCloud platform.

## Luminescence screening assays

The reporter of *hutH-lux*, *hutG-lux*, *hutU-lux*, and *hutF-lux* was electroporated into *P. aeruginosa* PAO1 WT, Δ*hutC* mutant, and complemented strains. Following overnight cultivation, normalized bacterial suspensions (OD$_{600}$ = 0.2) in fresh LB were supplemented with 100 μL medium to establish a 1:20 dilution ratio. Aliquots (200 μL) were transferred to 96-well round-bottom plates with clear bottoms and overlaid with 60 μL sterile mineral oil to prevent solvent evaporation. Bioluminescence (counts per second, CPS) and growth kinetics (OD$_{600}$) were monitored hourly for 24 h using a Synergy 2 multimode reader (TECAN). Transcriptional activity was quantified as normalized luminescence units (CPS/OD$_{600}$ ratio) throughout the growth phase.

## Western blot

Overnight bacterial cultures in LB broth were diluted 1:100 in fresh medium and grown to mid-log phase (OD$_{600}$ = 0.8). Cell suspensions (1 mL aliquots) were harvested via centrifugation at 12,000 × $g$ for 2 min, and pellets reconstituted in 60 μL SDS sample buffer. Equal protein quantities from lysates were resolved on 12% SDS-polyacrylamide gels under denaturing conditions. Following electrophoretic separation, proteins were electrotransferred onto PVDF membranes using a semi-dry transfer system. Membranes underwent blocking with 5% skim milk in TBST (10 mM Tris-HCl, 150 mM NaCl, and 0.05% Tween-20, pH 7.5) for 2 h at 25°C before overnight incubation at 4°C with target-specific primary antibodies. After three TBST washes (10 min each), blots were probed with HRP-linked secondary antibodies (1:5,000 dilution) for 60 min at ambient temperature. Chemiluminescent signal development was performed using the ECL Plus substrate (EpiZyme Biotech) according to the manufacturer's protocols.

## Protein expression and purification

For HutC protein production, recombinant *E. coli* BL21 (DE3) harboring the pET28a-*hutC* plasmid was cultured in LB broth at 37°C. At an OD$_{600}$ of 0.6, induction with 0.5 mM IPTG was initiated, followed by 20-h incubation at 18°C. Harvested cells were resuspended in lysis buffer (10 mM Tris-HCl, pH 7.5, 500 mM NaCl) and disrupted by sonication (Xinzhi, Ningbo, China). The clarified lysate was filtered (0.45 μm) and loaded onto a Ni-NTA affinity column (Qiagen, Germany). Sequential washing with 10% elution buffer (identical base components with 300 mM imidazole) preceded protein elution via a 50 mL linear gradient (10–100% elution buffer). Pooled target fractions were further purified using a HiLoad 16/60 Superdex size-exclusion column (GE Healthcare) pre-equilibrated with lysis buffer. SDS-PAGE analysis confirmed protein homogeneity. Final samples were concentrated with 30 kDa molecular weight-cutoff filters (Millipore) and cryopreserved at −80°C.

## Phenazine analysis

Configured standard solutions of PCA and 1-HP at 200 µM, and performed full-wavelength scanning using a microplate reader to determine their absorption peaks. Simultaneously, the supernatants of the WT PAO1, Δ*hutC*, and the complemented strain (normalized to the same $OD_{600}$) were performed full-wavelength scanning using a microplate reader. By comparing the maximum absorption peaks between the standard compounds and strains supernatant, it was determined that which specific phenazine compound was present in the Δ*hutC* mutant supernatant.

## High-performance liquid phase (HPLC)

To further extract and determine the dark brown substance in Δ*hutC* mutant supernatant, 180 µL of fermentation culture was mixed with 20 µL of diluted HCl (6 mol/L), and then, 540 µL chloroform was added with vigorous shaking for 2 min. After centrifugation at 12,000 rpm for 5 min, 300 µL of the organic phase was taken into the new EP tube and an equal volume of methanol:chloroform (9:1, v/v) was added. Finally, solvents were completely removed using a Nitrogen Blowing Concentrator.

The absorbance at 245 nm was monitored by HPLC. The mobile phase was acetonitrile and water containing 0.1% TFA. HPLC analysis (Agilent C18 250 × 4.6 mm, 5 µm) under the following conditions: injection volume 20 µL; column temperature 28°C; and flow rate 0.7 mL/min.

## Electrophoretic mobility shift assay (EMSA)

EMSA was modified based on previous reports (17). Briefly, purified HutC protein was mixed with 2.0 ng/µL DNA probes (Table S4) in a total volume of 20 µL of gel shift loading buffer, which included 10 mM Tris-HCl (pH 7.5), 100 mM NaCl, 1 mM dithiothreitol, 3 ng/µL sheared salmon sperm DNA, and 1% glycerol. After incubation for 20 min at room temperature, the reaction mixtures were run on 6% non-denaturing acrylamide gel in 0.5 × Tris-Borate-EDTA buffer at 90 V. The gels were subsequently stained with GelRed staining solution (TransGen Biotech) and visualized under UV light using a Bio-Rad transilluminator.

## Statistical analysis

Each experiment was carried out in triplicate and three independent biological replicates. To assess data significance, one-way ANOVA coupled with Tukey's multiple comparisons test was employed for intergroup comparisons. Quantitative findings were expressed as mean values ± standard deviation (SD) from biological triplicates. A probability threshold of $P < 0.05$ was established as the criterion for statistical significance in all comparative analyses.

## RESULTS

### Comprehensive transcriptional analysis of HutC in *P. aeruginosa*

Bacterial transcription regulation is one of the main mechanisms by which bacteria rapidly reproduce and adapt to the complex external environments. Many studies have reported that GntR family regulators can participate in the regulation of many cellular and biological activities, such as *P. aeruginosa*, *Vibrio parahaemolyticus*, *Brucella abortus*, and *Pseudomonas fluorescens* SBW25 (11, 17–21). HutC belongs to a member of the GntR family, also has the similar effects and functions of the GntR family. Blast analysis indicated that HutC shows homology to other GntR family genes (Fig. S1). In order to unlock the biological function of the HutC regulon in *P. aeruginosa*, we carried out a comprehensive transcriptome analysis between the WT PAO1 and the Δ*hutC* mutant by means of RNA-seq. The results of transcriptome analysis indicated that 844 genes had alterations at the transcriptional level (>2-fold, $P < 0.05$), among which 310 genes were upregulated and 534 genes were downregulated in the Δ*hutC* mutant

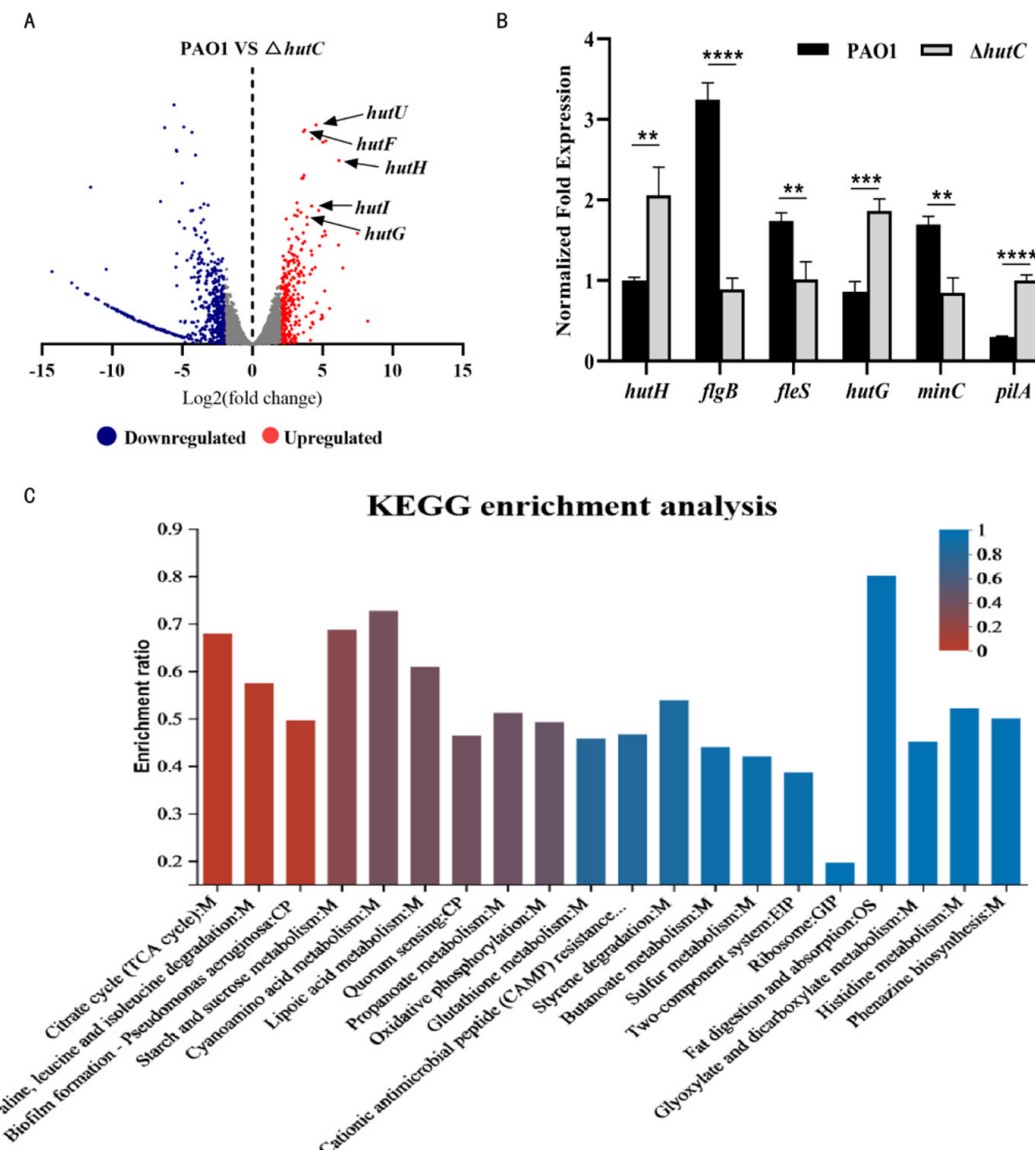

**FIG 1** Transcriptome analysis identified HutC-regulated genes in *P. aeruginosa*. (A) Volcano plots of the DEGs were analyzed between the WT PAO1 and Δ*hutC* strain by RNA-seq. Red dots, upregulated. Green dots, downregulated. FC, fold change. Genes involved in histidine catabolism are presented. (B) qRT-PCR analysis of six representative genes was evaluated for validation of the transcriptome data in WT PAO1 and Δ*hutC* strain. Error bars indicate the mean ± SD of three independent experiments. Statistical significance was determined by unpaired *t*-tests. **$P < 0.01$; ***$P < 0.001$; and ****$P < 0.0001$. (C) KEGG pathway enrichment analysis classification, top 20 pathway enrichment were given.

(Fig. 1A; Table S1). To comprehensively analyze the biological processes and pivotal pathways that the differentially expressed genes were predominantly involved in, we performed the KEGG pathway enrichment. KEGG analysis results showed that several pathways were enriched, including TCA cycle, biofilm formation, two-component system, histidine metabolism, and phenazine biosynthesis (Fig. 1C). Also, *hut* operon genes were significantly upregulated in RNA sequencing, with *hutH, I, G, U, F* (required for histidine catabolism) as highly (13- to 72-fold) upregulated genes in the Δ*hutC* strain compared with the WT PAO1 (Fig. 1A; Table S1). To verify the reliability of the RNA-seq data, we randomly selected six genes and their expression was analyzed using qRT-PCR. The expression of these selected genes was regulated by *hutC* (Fig. 1B). These results were consistent with the transcriptome data.

## Characterization of the gene clusters involved in histidine catabolism

Histidine (chemical formula $C_6H_9N_3O_2$) plays pivotal roles in the metabolic network of *P. aeruginosa*, serving as an essential source of carbon, nitrogen, and energy. The degradation product of histidine, urocanate (chemical formula $C_5H_4N_4O_3$), is the initial intermediate in the histidine degradation pathway. Remarkably, urocanate retains nearly all the carbon and nitrogen source derived from its precursor. Interestingly, *P. putida* encounters a significant limitation growing in a medium with urocanate as the sole substrate, due to the inherent inability to directly assimilate exogenous urocanate (22). However, *P. aeruginosa* harbors the putative urocanate transporter gene; the exogenous urocanate can be assimilated and absorbed (23). Based on homology analysis (Fig. S2), the *hut* operon genes in *P. aeruginosa* PAO1 include *hutH* (PA5098), *hutI* (PA5092), *hutG* (PA5093), *hutU* (PA5100), and *hutF* (PA5106), all of which are involved in histidine catabolism. To assess the roles of the *hut* operon in histidine catabolism, we generated six mutant strains including Δ*hutC,* Δ*hutH,* Δ*hutI,* Δ*hutG,* Δ*hutU,* Δ*hutF* and their corresponding complemented plasmids, and then observed their growth by adding the corresponding substrate as the sole carbon and nitrogen source in M9 medium. The growth curves showed that the six deletion strains exhibited the same growth patterns as the WT PAO1 in M9 medium plus glucose (Fig. 2). However, there were marked growth impairments when histidine or urocanate served as the sole carbon and nitrogen source, respectively (Fig. 2). Furthermore, the survival phenotype of these mutants could be restored back to the WT levels following the introduction of the corresponding complemented plasmid. These results suggest that HutC, HutH, HutI, HutG, HutU, and HutF play critical roles in histidine catabolism.

## HutC regulates the *hut* operon

Transcriptome profiling revealed a significant upregulation of *hutH*, *hutI*, *hutG*, *hutU*, and *hutF* mRNA levels in the Δ*hutC* mutant strain compared to in the wild-type strain (Fig. 1A; Table S1). To evaluate this regulation of the *hut* operon by *hutC* in more detail, four promoter-*lux* fusions plasmids were constructed including *hutH-lux*, *hutU-lux*, *hutF-lux*, and *hutG-lux* (*hutG* shares the same promoter with *hutI*). Then, we quantitatively assessed their promoter activity in WT PAO1 strain, the Δ*hutC* mutant, and its complemented strain (Δ*hutC*/p-*hutC*). The results showed that these fused promoters displayed a marked upregulation activity in the Δ*hutC* mutants (Fig. 3A; Fig. S3A), which was in alignment with the qRT-PCR data (Fig. 1C). To further validate this result, we used western blot to verify the *hutG* protein level in WT PAO1, Δ*hutC* mutant with the complemented strain. Consistent with the transcription analysis, the protein level of HutG was significantly elevated in the Δ*hutC* mutant compared to both WT PAO1 and the complemented strain ($P < 0.01$) (Fig. 3B). Altogether, these findings demonstrate that HutC as a negative regulator effectively controls the expression of *hut* operon in *P. aeruginosa*.

To elucidate the regulatory role of HutC on the expression of *hut* operon, we conducted a series of EMSAs by detecting the interaction between the purified recombinant HutC protein (Fig. 3B) and the intergenic regions of *hutH*, *hutU*, *hutF*, and *hutIG* promoters. EMSA experiments showed that HutC can efficiently bind only to the *hutIG* promoter region (Fig. 3C), but cannot bind to *hutH*, *hutU*, and *hutF*. To further ascertain the exact DNA sequence by HutC recognition, we conducted additional EMSAs using various truncated fragments of *hutIG* promoter region. EMSAs showed that HutC was bound to the adjacent sites (Site: <u>CGGCGGC</u>CTGA<u>GCCGCCG</u>, −108 to −90) located upstream of the ATG start codon of *hutIG* (Fig. 3D; Fig. S3C). Collectively, these findings demonstrate that HutC directly represses the expression of genes encoding key enzymes involved in histidine catabolism.

## Non-targeted metabolomics analysis of HutC in *P. aeruginosa*

In a previous experiment, we found that knocking out *hutC* exhibited a dramatic accumulation of dark brown substance compared with the WT PAO1 in LB medium

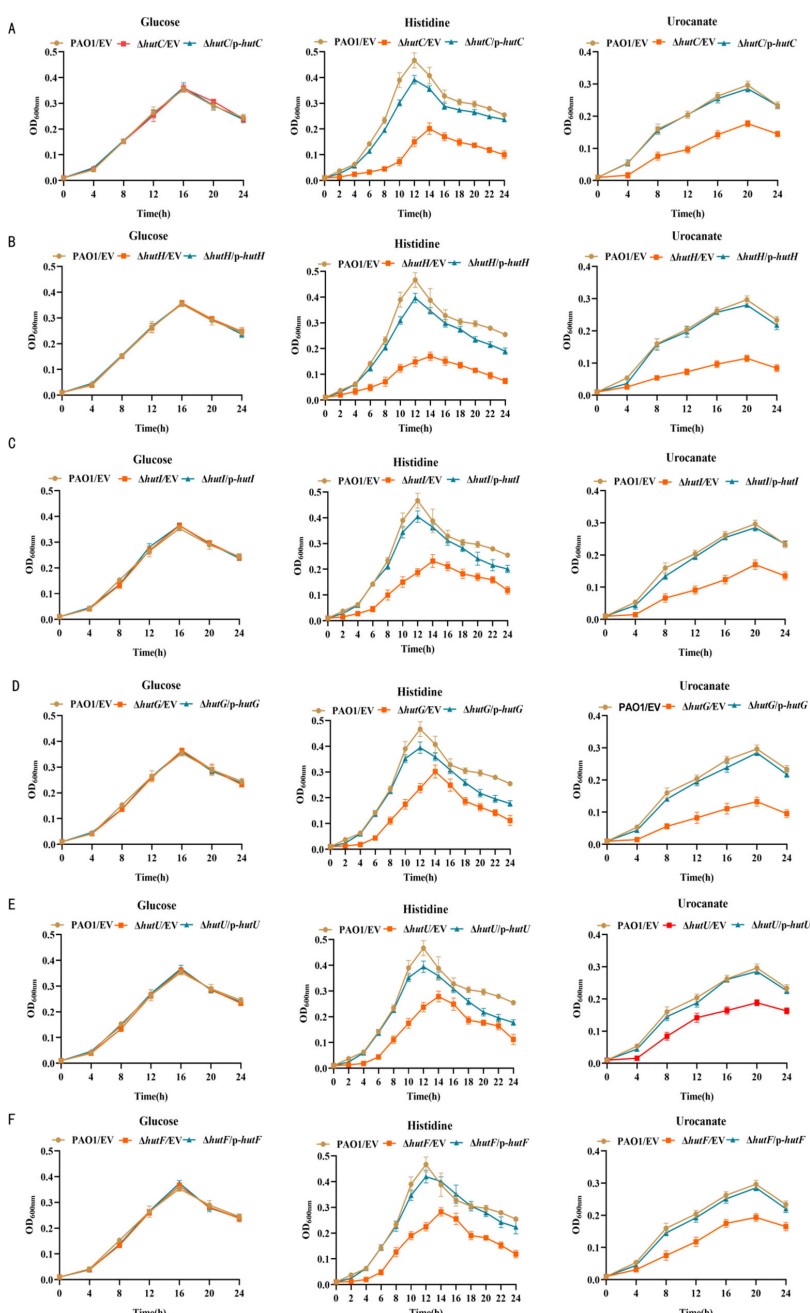

**FIG 2** Growth phenotypes of the *P. aeruginosa* histidine catabolic deletion mutants. (A–F) Carbon and nitrogen utilization phenotype assays were conducted on the WT PAO1, Δ*hutC,* Δ*hutH,* Δ*hutI,* Δ*hutG,* Δ*hutU,* Δ*hutF,* and their complemented strains using minimal medium supplemented with 10 mM glucose, histidine, or urocanate as the sole carbon and nitrogen source. Complementation of these mutants by introducing the corresponding complemented plasmid restored wild-type phenotype. EV represents an empty vector. Error bars indicate the SD of the mean from three independent experiments.

(Fig. 4A). To identify the composition of this brown substance, we performed non-target metabolomic analysis to determine metabolite differences by liquid chromatography mass spectrometry. In total, 966 metabolites were identified with chemical names and structures (Table S2). The OPLS-DA score plots demonstrated a distinct differentiation between the two groups, accounting for 52.7% of the variance and the second principal component (PC2) contributing 11.9% (Fig. 4B). KEGG pathway analysis showed that

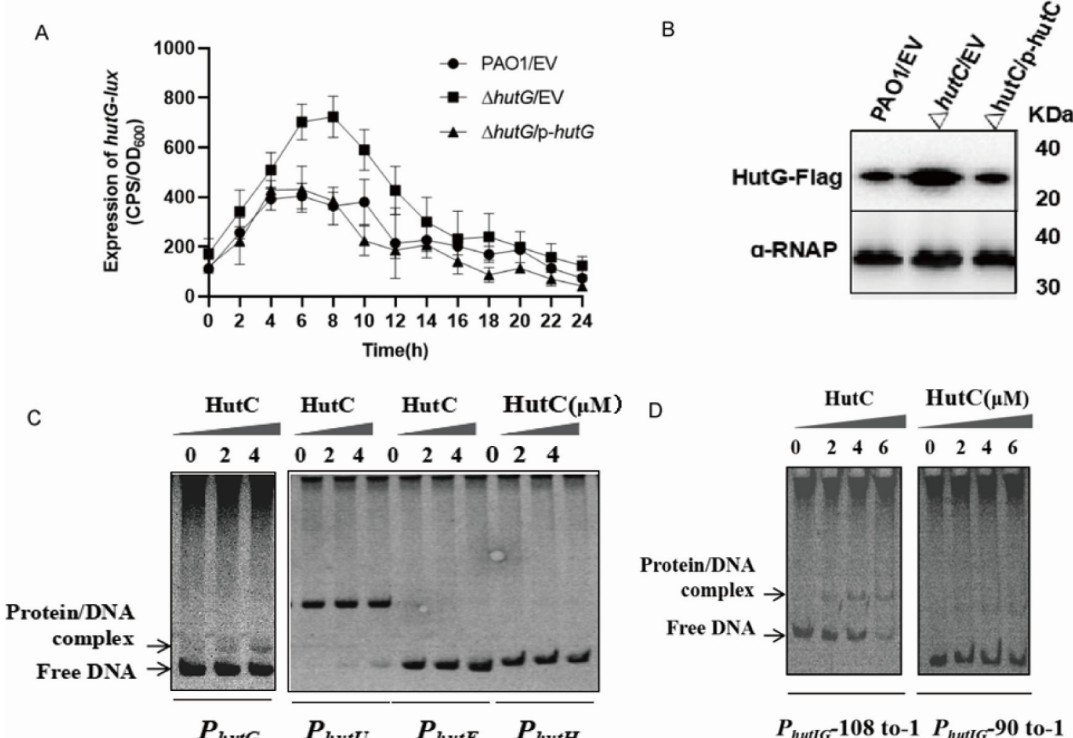

**FIG 3** HutC is involved in the regulation of the histidine catabolism directly. (A) The promoter activity of *hutG* was measured in the WT PAO1, Δ*hutC*, and the complemented strain cultured in LB medium. Error bars indicate the mean ± SD of three independent experiments. EV represents the empty vector. (B) A mini-CTX plasmid for the expression of HutG-Flag chimaera driven by their promoters was integrated into *P. aeruginosa* derivative strains. The tagged proteins were detected using a Flag antibody. Immunoblots for α-RNAP (RNA polymerase α) served as a loading control. (C) EMSAs showing that HutC could only bind to the promoter region of hutG but not to *hutH*, *hutU*, and *hutF*. Each reaction mixture contains 2.0 ng/µL of *hutG*/DNA. The concentration of HutC was indicated above its lane. (D) EMSAs showing that HutC binds to the promoter region of $hutG_{-108 \text{ to } -1}$, but not $hutG_{-90 \text{ to } -1}$. Data are representative of three independent replicates.

the metabolites were predominantly enriched in amino acid metabolism (e.g., histidine, phenylalanine, arginine, and proline metabolism), nucleotide metabolism (purine and pyrimidine metabolism), secondary metabolite biosynthesis (pantothenic acid, coenzyme A, niacin, and nicotinamide biosynthesis) and signal transduction (two-component system) (Fig. 4C). Most of the metabolites identified are amino acids, carboxylic acids, nucleotides, cofactors, and vitamins (Fig. 4E). The data showed that 167 metabolites were differentially expressed between the WT PAO1 and Δ*hutC* mutant, while 102 were upregulated and 65 were downregulated significantly (>2-fold, $P < 0.05$). In addition, significantly different metabolites mainly include phenazine compounds (PCA, phenazine-1-carboxylic acid; 1-HP, 1-hydroxyphenazine; and PYO, pyocyanin) (Fig. 4D).

## Phenazines appear to be the effector for HutC

HutC, as a typical GntR family regulator, possesses a dual-domain structure with a DNA-binding N-terminal domain (NTD) and an effector-binding C-terminal domain (CTD). Its regulation of gene expression depends on the binding of an effector molecule in response to environmental signals such as the availability of different carbon sources (24, 25). Previous studies have reported that phenazines are signaling molecules for bacterial recognition (8, 26, 27). Moreover, phenazine compounds imparted different pigments to *P. aeruginosa* cultures, and the Δ*hutC* mutants produced a large amount of dark brown substance (Fig. 4A). In addition, phenazine compounds were also identified through metabolomic analysis in our study (Fig. 4D). Then, we further determined the brown substance in the Δ*hutC* strain supernatant through full-wavelength scanning. The

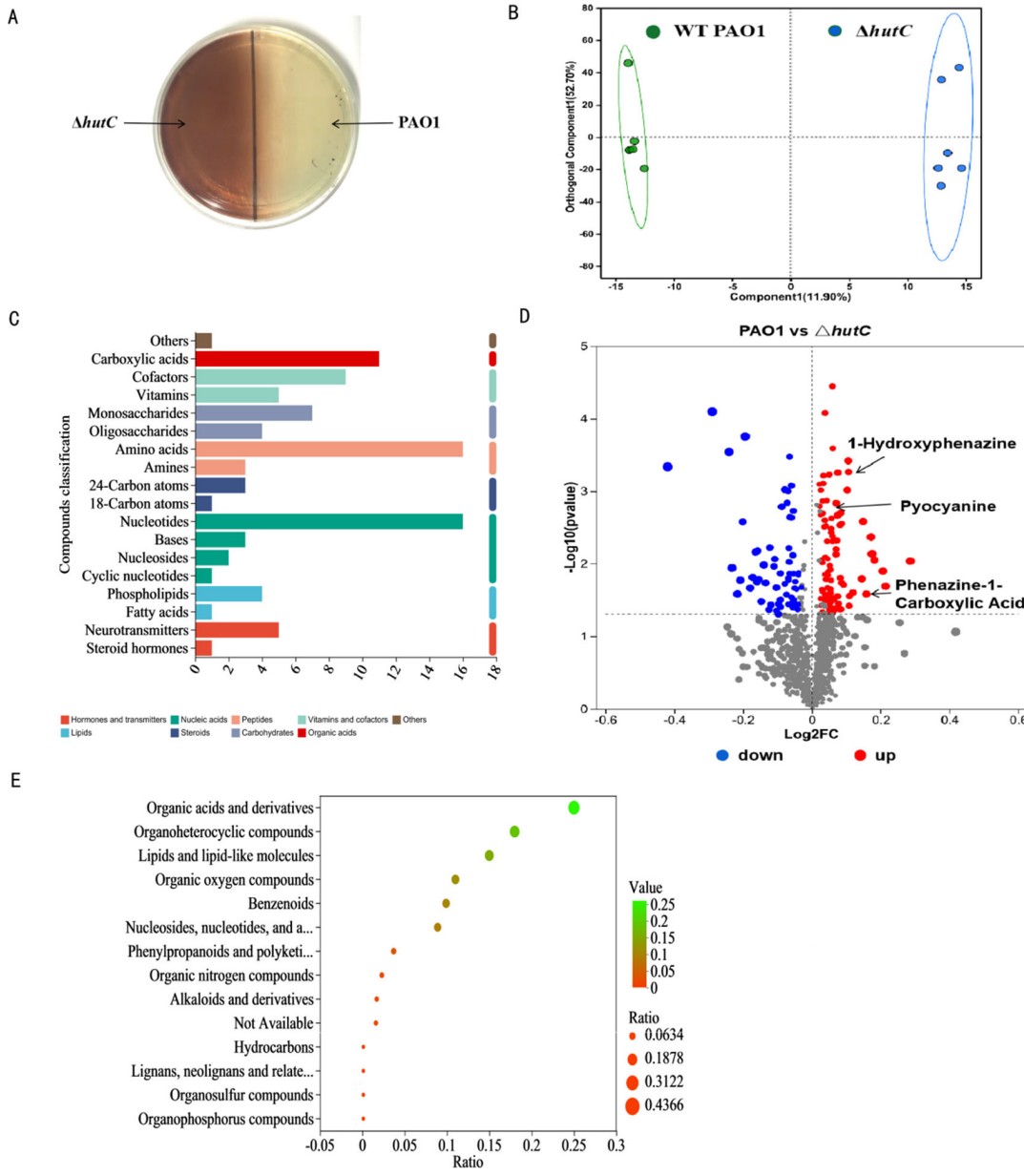

**FIG 4** Metabolomic analysis identified HutC-regulated genes in *P. aeruginosa*. (A) The brown substance production of the WT PAO1, Δ*hutC* mutant, and the complemented strain was detected in LB broth for 24 h. (B) Pancreatic metabolic profile significantly differed between WT PAO1 and Δ*hutC* group by orthogonal least partial squares discriminant analysis (OPLS-DA) method. Test for the OPLS-DA model showed that the model for this study was valid. (C) The analysis of KEGG enrichment pathway in WT PAO1 group compared with Δ*hutC* group. (D). Metabolites with significant differences between the WT PAO1 group and the Δ*hutC* groups were presented in the volcano plot. The red points represented the upregulated metabolites, and the blue points represented the downregulated metabolites. Compounds involved in histidine catabolism are presented. (E) HMDB compound classification of bacterial sedimentation metabolites between WT PAO1 group and Δ*hutC* groups. All compounds were divided into 10 clades according to the HMDB superclass.

results demonstrated that the standard PCA exhibited a prominent absorption peak at 380 nm. Similarly, the Δ*hutC* strain supernatant also showed a distinct absorption peak at this wavelength, whereas neither the WT PAO1 strain nor the complementary strain displayed significant peaks (Fig. 5A). Moreover, further experimental validation was obtained through HPLC. The position of this peak in extracts from the Δ*hutC* mutant strain is consistent with that of standard PCA (Fig. S4). These results strongly corroborate the findings from metabolomic analysis, confirming that the Δ*hutC* strain specifically secretes PCA as its primary phenazine metabolite.

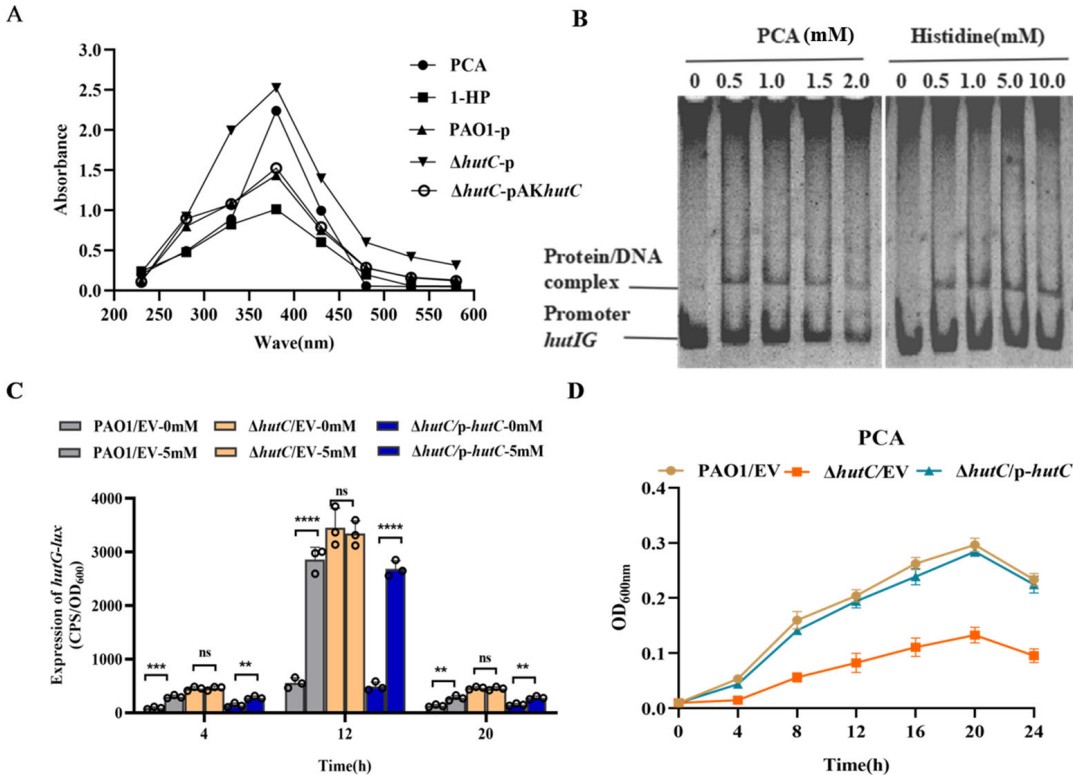

**FIG 5** Determining the effectors of HutC. (A) The absorption peaks of standard phenazine-1-carboxylic acid (PCA), 1-hydroxyphenazine (1-HP), and the culture supernatant of WT PAO1, ΔhutC, and the complemented strain were measured by full-wavelength scanning using a microplate reader. The concentration of PCA and 1-HP was 200 µM. (B) EMSA showing the effect of different concentrations of PCA and histidine on the DNA-binding ability of HutC-His₆. Each reaction mixture contains a constant 2 ng/µL of hutIG/DNA, 4 µM HutC protein, and increasing concentrations of small molecules as indicated, respectively. (C) Expression of the hutG-lux reporter in the absence and the presence of 5 mM PCA was measured in WT PAO1, ΔhutC, and the complemented strain, respectively. Data from n = 3 biological replicates are reported as mean ± SD (ns. not significant; **P < 0.01; ***P < 0.001; ****P < 0.001, Student's t-test). The strains contain either the empty vector pAK1900 or the complemented plasmid (D). The growth assays were conducted on the WT PAO1, ΔhutC, and its complemented strains using minimal medium supplemented with 10 mM PCA as the sole carbon and nitrogen source. Complementation of these mutants by introducing the corresponding complemented plasmid restored wild-type phenotype. EV represents an empty vector. Error bars indicate the SD of the mean from three independent experiments.

Based on previous studies (17), we speculated that PCA may be a potential effector of HutC. To verify this conjecture, we performed EMSAs to assess the impact of these compounds on the binding affinity of HutC to its targets. The addition of 2 mM PCA weakened the binding of HutC to the hutG promoter. In contrast, the addition of 10 mM histidine did not produce a similar effect (Fig. 5B). So we reasoned that the binding of PCA to HutC would affect HutC-mediated regulation of gene expression. To verify this, we examined the transcription activity of hutG-lux in the ΔhutC mutant strain in the presence or absence of PCA compared to WT PAO1. We observed that PCA strongly induced the activity of hutG in WT PAO1 and the complemented strain. There was no apparent hutG gene expression difference in the ΔhutC mutant strain (Fig. 5C). We also measured the growth of the ΔhutC mutant strain in M9 medium supplemented with PCA and 1-HP substrate as the sole carbon and nitrogen source, respectively. Our data showed that the ΔhutC mutant grew slower than WT PAO1 in a medium with PCA (Fig. 5D), but grew like WT PAO1 in a medium with 1-HP (Fig. S5). Taken together, these results suggest that PCA is involved in HutC-mediated regulation of gene expression.

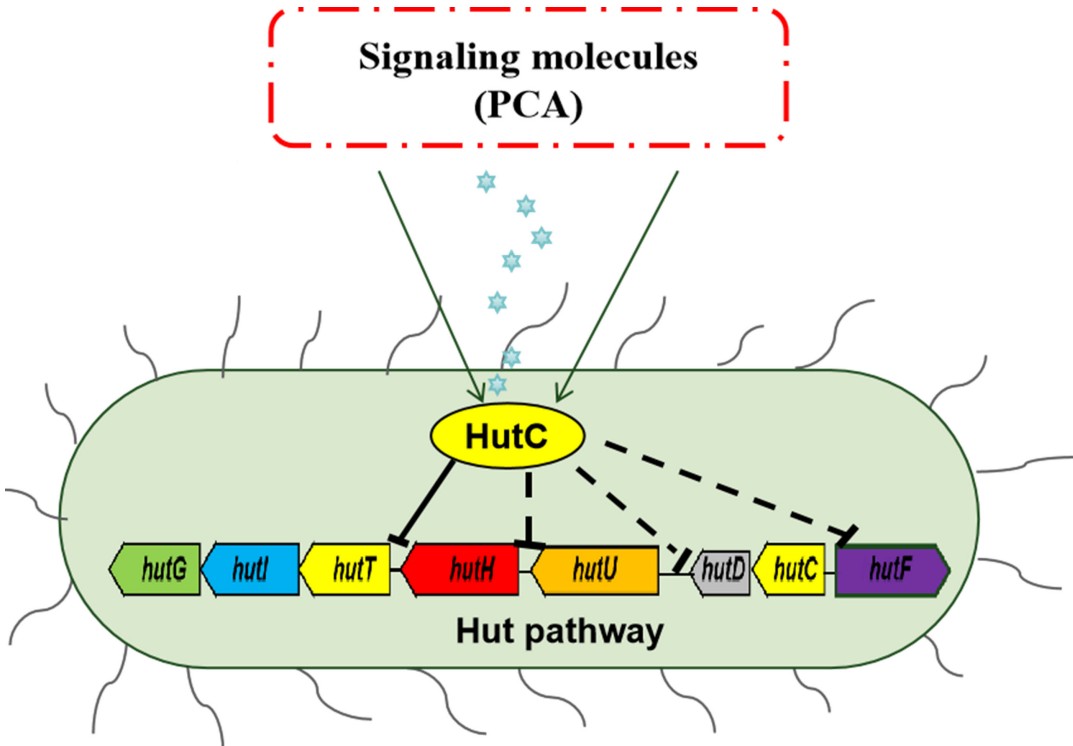

**FIG 6** Proposed model of HutC regulation of histidine catabolism in *P. aeruginosa*. HutC directly represses the expression of the *hutIG* operon. HutC also indirectly negatively regulates the expression of *hutH*, *hutU,* and *hutF*, key enzymes of the histidine pathway. HutC-mediated inhibition can be relieved by signal molecules, such as PCA and 1-HP. T-bars present negative regulation; solid and dotted lines indicate direct and indirect regulation, respectively.

## DISCUSSION

The ability of some opportunistic pathogens to adapt smoothly to various environments is most likely a main cause of the associated diseases. Amino acids, as nutrient sources, are important for the proliferation and metabolic processes of many bacteria (28). Histidine is one of the most energetically expensive amino acids to biosynthesize, and thus both the import/uptake and degradation/secretion of histidine are tightly regulated to maintain homeostatic levels (10). In this study, we coupled reverse genetics with bioinformatics to identify a *hut* operon that participates in histidine catabolism in *P. aeruginosa* (Fig. 2; Fig. S2). PA5098 (*hutH*) liberates the amino group of histidine and produces the first intermediate of urocanate (Fig. S2 and S4). Then PA5100 (*hutU*) hydrates urocanate to imidazolone propionate (IP), which is cleaved to formiminoglutamate (FIG) by PA5092 (*hutI*) (Fig. S2 and S4). In bacteria, there are two different pathways for FIG. In some bacteria, FIG is enzymatically hydrolyzed to formamide and glutamate. In other bacteria, such as certain *Pseudomonas* species, FIG is converted to formate and glutamate through the action of PA5106 (*hutF*) and PA5093 (*hutG*) (Fig. S2 and S4) (3, 29–31). Importantly, our data identified five enzymes involved in the histidine catabolism in *P. aeruginosa*, and this has also been confirmed by bioinformatics analysis. Strains with deletions of *hutCUHIGF* cannot grow well on histidine as the sole carbon and nitrogen source (Fig. 2); especially, the Δ*hutH* mutant grew relatively more slowly. Strains lacking HutH, which is the first enzyme of histidine catabolism, are defective for histidine as carbon and nitrogen source. Interestingly, exogenous urocanate cannot be directly absorbed and utilized by cells. Some bacteria, such as *P. putida*, cannot utilize urocanate for growth due to lacking the urocanate-specific transporter (23). In contrast, *P. aeruginosa* and *Klebsiella pneumoniae* harboring a urocanate transporter gene were thus able to utilize urocanate (32). Our findings also indicated that the deletion of *hutCUHIGF* was unable to efficiently utilize urocanate for growth (Fig. 2).

Hut system is subject to an intricate system of regulation, where HutC is a major transcriptional regulator. In this study, we found that HutC represses the *hut* pathway directly, which was confirmed from the results of RT-qPCR and promoter activity assay (Fig. 1B and 3A; Fig. S3A). Although HutC is located between the divergently transcribed *hutF* and *hutHUIG* operons and could repress both operons (22), our results revealed that HutC bound to the operator regions of the *hutIG* operon directly and not *hutF* operon by EMSA (Fig. 3C). Previous studies have shown that HutC binds to the region of the *hutU* promoter in *P. fluorescens* and in *P. putida* (11, 22). The possible reason is that the *hut* cluster from *hutU-G* forms a single operon in the *P. fluorescens*, whereas *hutIG* has its own promoter and is transcribed as an independent unit in the *P. aeruginosa* (10). More specifically, HutC affects the rate of histidine decomposition by directly targeting the promoter DNA of *hutIG*. It is worth noting that the binding sites of the HutC and *hutIG* gene found in *P. aeruginosa* show little sequence similarity with the HutC consensus sequence previously identified in other bacteria (3, 7, 11). This suggests that HutC may have evolved specific binding to distinct DNA sites conferring new functions to adapt to sophisticated environments (33).

Bacteria responding to extracellular signaling molecules to control microbial metabolism and virulence are prevalent among diverse bacterial species (34–36). HutC, as a representative member of the GntR family, possesses the typical two-domain proteins with an N-terminal DNA-binding and a C-terminal ligand-binding domain (37). Here, we found that HutC sensed PCA as its effector to regulate the expression of *hut* operon (Fig. 5B). The addition of 2 mM PCA could directly prevent the binding of HutC to *hutIG* promoter DNA. However, 10 mM histidine did not affect the binding of HutC to *hutIG*. The reason may be that the histidase (HutH), the first enzyme of the *hut* pathway, is necessary to produce urocanate to relieve repression of the *hut* operons via the interactions between urocanate and HutC when histidine content increases in the environment. Consequently, histidine is not the most suitable signaling molecule that activates the *hut* operon. Phenazines, as redox-active signaling molecules, participate in various physiological and pathological processes in various bacterial species. These molecules can change the cellular oxidation-reduction balance and play a signaling function through regulating gene expression, contributing to the bacteria fitness and pathogenicity (27, 38). For example, PYO as a signal molecule influences the intracellular redox state through interfering with central metabolic pathways in *P. aeruginosa* (39). In this study, phenotypic identification of HutC and EMSA were combined with metabolomic analyze to comprehensively elucidate this mechanism (Fig. 4A, D, 5A and B). Overall, we propose a new model in which HutC responds to phenazine metabolites and regulates the *hut* system to enhance ecological adaptability of *P. aeruginosa* (Fig. 6).

In summary, our findings offer a biochemical elucidation of the transcriptional regulation mechanisms of HutC which modulates gene expression of *hut* operon in response to PCA ligand. The findings highlight PCA as a novel signaling molecule that is critical for HutC-mediated histidine catabolism. Further studies will elucidate how PCA interacts with HutC and whether it is involved in other regulatory pathways. Moreover, the binding of HutC to PCA could be suggested as a biosensor or potential therapeutic target for diagnosis and treatment of infectious diseases caused by *P. aeruginosa*.

## ACKNOWLEDGMENTS

The authors acknowledge Illumina sequencing support from Majorbio Biomedical Technology Co., Ltd. and Majorbio cloud platform.

This work was supported by the research foundation for the Doctoral Program of Chang Zhi Medical College (grant number: 2024BS11).

## AUTHOR AFFILIATIONS

[1]Department of Microbiology, Changzhi Medical College, Changzhi, Shanxi, China
[2]High Latitude Crops Institute, Shanxi Agricultural University, Datong, Shanxi, China

³Department of Biochemistry, Changzhi Medical College, Changzhi, Shanxi, China
⁴Department of Basic Medicine, Changzhi Medical College, Changzhi, Shanxi, China

## AUTHOR ORCIDs

Guoyan Cui  http://orcid.org/0000-0003-0599-828X
Jia Cui  http://orcid.org/0000-0003-1362-8609

## FUNDING

| Funder | Grant(s) | Author(s) |
|---|---|---|
| Ministry of Education, Science and Technology Development Center | 2024BS11, 20251112 | Guoyan Cui |

## AUTHOR CONTRIBUTIONS

Guoyan Cui, Conceptualization, Data curation, Funding acquisition, Investigation, Methodology, Resources, Software, Supervision, Validation, Visualization, Writing – original draft, Writing – review and editing | Zhuang Li, Conceptualization, Data curation, Investigation, Software, Validation, Visualization, Writing – original draft | Yiqiang Zhang, Data curation, Investigation, Methodology, Supervision, Validation, Writing – original draft | Fengping Wang, Data curation, Investigation, Methodology, Supervision | Dan Li, Data curation, Investigation, Methodology, Supervision | Yujiao Zhao, Data curation, Investigation, Methodology, Supervision | Jia Cui, Conceptualization, Data curation, Investigation, Methodology, Supervision, Validation, Writing – original draft

## ADDITIONAL FILES

The following material is available online.

### Supplemental Material

**Supplemental figures (Spectrum00816-25-s0001.docx).** Fig. S1 to S6.
**Supplemental tables (Spectrum00816-25-s0002.docx).** Tables S1 to S4.

### Open Peer Review

**PEER REVIEW HISTORY (review-history.pdf).** An accounting of the reviewer comments and feedback.

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
