## [Reviewer comments · Microbiology Spectrum]

Microbiology Spectrum

A GntR family regulator HutC senses PCA to regulate histidine catabolism in *Pseudomonas aeruginosa*

Guoyan Cui, Li Zhuang, Yiqiang Zhang, Fengping Wang, Dan Li, Yujiao Zhao, Jia Cui

Corresponding Author(s): Guoyan Cui, Changzhi Medical College

Review Timeline:

Submission Date:	April 4, 2025
Editorial Decision:	May 18, 2025
Revision Received:	September 8, 2025
Editorial Decision:	October 3, 2025
Revision Received:	October 28, 2025
Accepted:	November 3, 2025

Editor: Monica Cartelle Gestal

Reviewer(s): Disclosure of reviewer identity is with reference to reviewer comments included in decision letter(s). The following individuals involved in review of your submission have agreed to reveal their identity: STEPHEN DELA Ahator (Reviewer #2); Paul Brown (Reviewer #3)

Transaction Report:

DOI: <https://doi.org/10.1128/spectrum.00816-25>

Re: Spectrum00816-25 (**A GntR family transcriptional repressor HutC sensing PCA regulated histidine catabolism in *Pseudomonas aeruginosa***)

Dear Dr. cui jia:

Thank you for the privilege of reviewing your work. Below you will find my comments, instructions from the Spectrum editorial office, and the reviewer comments.

This manuscript investigates the role of HutC in histidine catabolism and the regulation of the Hut operon. The authors apply omics approaches to investigate the function of GntR in *Pseudomonas aeruginosa*. While the manuscript has potential there are some concerns that need to be addressed especially regarding grammar and clarity. Please use the comments of both reviewers as a guidance for the modifications, but bare in mind that reviewer 1 is a non-detailed list of clarity issues, so try to apply changes to the totally of the document.

Revision Guidelines

Sincerely,
Monica Cartelle Gestal
Editor
Microbiology Spectrum

Reviewer #1 (Comments for the Author):

This is a potentially interesting paper by Yan and colleagues about the role of HutC in histidine catabolism. However, the paper is extremely difficult to read and follow, and I struggled to ascertain what exactly the authors were showing and how to interpret the data at several points. A non-exhaustive list of issues:

1. The title is poor, requiring readers to know what a GntR family repressor is.
2. There are many instances throughout where the grammar is incorrect. There are also lines (220-252) where individual words are split across two lines, which is extremely difficult to read.
3. Figure 1A - the choice of red and green would make the volcano plot uninterpretable for someone with red/green colorblindness
4. Figure 2 - the symbols are too small, as is the text, and it is difficult to figure out which lines are which strains. This would be a good location for color!
5. Figure 3 - panel A does not show promoter activities. (In fact it is duplicated from Figure 2)
6. Figure 3C - it seems like there is a gel shift with no added protein? Also, which is the tested hutU promoter so much longer than the others?
7. Figure 4A - the legend talks about three strains but only two are shown.
8. Figure 4B legend. What is "Pancreatic metabolic profile"???
9. Figure 4C - the text is way too small
10. Figure 5. The text is appropriately circumspect about the potential role of PCA but the title of the figure is overly definitive. It's possible that PCA is one contributor to HutC regulation.

Reviewer #2 (Comments for the Author):

The work by Cui and colleagues, presents a study on the transcriptional regulation of the hut operon by HutC, a GntR family transcriptional repressor, in *Pseudomonas aeruginosa*. They apply Transcriptomics, gene deletion and complementation, EMSA, metabolomics and functional assays to elucidate the regulatory mechanisms of hutC in *Pseudomonas*. They show the role of hutC in histidine catabolism and its interaction with phenazine-1-carboxylic acid (PCA) as a regulatory ligand

Comments and Suggestions.

There are a few errors in grammar, sentence structures and word choice which can be corrected to improve clarity.

From the transcriptomic, what was the expression of the urocanate transporter, and how does it explain the growth patterns of the Wild type and the hut mutants in urocanate?

What negative control was used for the EMSA binding assay for assessing PCA disruption of HutC binding?

Did exogenous addition of PCA and 1-HP affect the production of the brown pigment in the hutC mutant or the WT? Did you observe any phenotypic characteristics in the wild type or hut mutants, such as changes in growth pattern, supporting the role of PCA and 1-HP in hutC pathway?

While it is stated that PCA disrupts HutC binding, the biochemical mechanism is speculative and not explored in detail, as there is less information on the structural binding site or conformation changes. Please provide more details on PCA production and its environmental triggers or conditions.

**A GntR family transcriptional repressor HutC sensing PCA**
**regulated histidine catabolism in *Pseudomonas aeruginosa***

Guoyan Cui^{1#}, Li Zhuang^{2#}, Yiqiang Zhang³, Fengping Wang⁴, Dan Li⁴, Yujiao Zhao⁴,
and Jia Cui^{1*}

¹*Department of Microbiology, Changzhi Medical College, Changzhi, Shanxi, China*

²*High Latitude Crops institute, Shanxi Agricultural University, Datong, Shanxi, China*

³*Department of Biochemistry, Changzhi Medical College, Changzhi, Shanxi, China*

⁴*Department of Basic Medicine, Changzhi Medical College, Changzhi, Shanxi, China*

*Correspondence: Jia Cui, Department of Microbiology, Changzhi Medical College,
Changzhi 046000, China. Email: cuijia@czmc.edu.cn

#These authors contributed equally to this work.

**Abstract**

[revised manuscript text omitted]

- 16. Mortensen, S. A., Rosenfeldt, F., Kumar, A., Dolliner, P., Filipiak, K. J., Pella, D.,
Alehagen, U., Steurer, G., Littarru, G. P., & Q-SYMBIO Study Investigators (2014).
The effect of coenzyme Q10 on morbidity and mortality in chronic heart failure:
results from Q-SYMBIO: a randomized double-blind trial. *JACC. Heart failure*, 2(6),
641–649.
- 17. Cui G.Y., Zhang Y.X., Xu X.X., LiuY.Y., Li Z., and Wu M., et al. (2022) PmiR
senses 2-methylisocitrate levels to regulate bacterial virulence in *Pseudomonas*
*aeruginosa*. *Science Advances*, 49:eadd4220.
- 18. Xu X.J., Yan Y.F., Huang J., Zhang Z.H., Wang Z., Wu M., and Liang, H.H.
(2022) Regulation of uric acid and glyoxylate metabolism by UgmR protein in
*Pseudomonas aeruginosa*. *Environ Microbiol*, 7, 3242-3255.
- 19. Daddaoua A., Corral-Lugo A., Ramos J.L., and Krell T. (2017) Identification of
GntR as regulator of the glucose metabolism in *Pseudomonas aeruginosa*. *Environ*
*Microbiol*, 9, 3721-3733.
- 20. Gu D., Meng H.M., Li Y., Ge H.J., and Jiao X.N. (2019) A GntR family
transcription factor (VPA1701) for swarming motility and colonization of *Vibrio*
*parahaemolyticus*. *Pathogens*, 4, 235.
- 21. Li Z.Q., Wang S.L., Zhang H., Zhang J.L., Xi L., Zhang J.B., and Chen C.F. (2017)
Transcriptional regulator GntR of *Brucella abortus* regulates cytotoxicity, induces the
secretion of inflammatory cytokines and affects expression of the type IV secretion
system and quorum sensing system in macrophages. *World J Microbiol Biotechnol*, 3,
60.
- 22. Allison S.L., Phillips A.T., (1990) Nucleotide sequence of the gene encoding the
repressor for the histidine utilization genes of *Pseudomonas putida*. *J Bacteriol* 172,
5470–5476.
- 23. Zhang X.X., Chang H., Tran S.L., Gauntlett J.C., Cook G.M and Rainey P.B (2012)
Variation in transport explains polymorphism of histidine and urocanate utilization in
a natural *Pseudomonas* population. *Environ Microbiol*, 14,1941-1951.

- 24. Wang Y., Cao Q., Cao Q., Gan J., Sun N., and Yang C., *et al.* (2021) Histamine
activates HinK to promote the virulence of *Pseudomonas aeruginosa*. *Sci Bull*
(Beijing), 66,1101-1118.
- 25. Yu H., Xiong J., Zhang R., Hu X.M., Qiu J., and Zhang D., *et al.* (2016) Ndk, a
novel host-responsive regulator, negatively regulates bacterial virulence through
quorum sensing in *Pseudomonas aeruginosa*. *Sci Rep*, 6, 28684.
- 26. Liang H.H., Duan J.L., Sibley C.D., Surette M.G., Duan K.M. (2011)
Identification of mutants with altered phenazine production in *Pseudomonas*
*aeruginosa*. *J Med Microbiol*, 60,22-34.
- 27. Sousa C.A., Marta R., Vale F., Simões M. (2024) Phenazines: Natural products for
microbial growth control. *HLife*, 3, 100-112.
- 28. Wu T., Wang X.Z., Dong Y., Xing C., Chen X.Y., and Li L., *et al.* (2022) Effects
of L-Serine on Macrolide Resistance in *Streptococcus suis*. *Microbiol Spectr*, 10,
e0068922.
- 29. Kaminskas E., Kimhi Y., Magasanik B. (1970) Urocanase and N-formimino-L-
glutamate formiminohydrolase of *Bacillus subtilis*, two enzymes of the histidine
degradation pathway. *J Biol Chem*, 245,3536-3544.
- 30. Magasanik B., Bowser H.R., (1955) The degradation of histidine by *Aerobacter*
*aerogenes*. *J Biol Chem*, 213,571-580.
- 31. Kendrick K.E., Wheelis M.L., (1982) Histidine dissimilation in *Streptomyces*
*coelicolor*. *J Gen Microbiol*, 128,2029-2040.
- 32. Schlesinger S., Magasanik B. (1965) Imidazolepropionate, a nonmetabolizable
inducer for the histidine-degrading enzymes in *Aerobacter aerogenes*. *J Biol Chem*,
24,4325-4330.
- 33. Nakagawa S., Gisselbrecht S.S., Rogers J.M., Hartl D.L. and Bulyk M.L. (2013)
DNA-binding specificity changes in the evolution of forkhead transcription factors.
*Proc Natl Acad Sci US A*, 110,12349-12354.
- 34. Xiao G., Zheng X.F., Li J.Y., Yang Y., Yang J., and Xiao N., *et al.* (2022)
Contribution of the EnvZ/OmpR two-component system to growth, virulence and

stress tolerance of colistin-resistant *Aeromonas hydrophila*. *Front Microbiol*,
13:,1032969.

35. Wang T.T., Qi Y.H., Wang Z.H., Zhao J.R., Ji L.X., and Li J., *et al.*, (2020)
Coordinated regulation of anthranilate metabolism and bacterial virulence by the
GntR family regulator MpaR in *Pseudomonas aeruginosa*. *Mol Microbiol*, 114, 857-
869 .

36. Rosenberg G., Yehezkel D., Hoffman D., Mattioli C.C., Fremder M., and Hadar
B.A., *et al.* (2021) Host succinate is an activation signal for *Salmonella* virulence
during intracellular infection. *Science*, 371,400-405.

37. Thanusha D.A., Inoka C.P. (2021) In silico characterization and virtual screening
of GntR/HutC Family transcriptional regulator MoyR: a potential monooxygenase
regulator in *Mycobacterium tuberculosis*. *Biology* (Basel), 10,1241.

38. Cátia A.S., Marta R., Francisca V., Francisca V., Manuel S. (2024) Phenazines:
Natural products for microbial growth control. *HLife*, 2,100-112.

39. Fang Y.L., Chen B., Zhou L., Jin K.M., Jiang H.X., He Y.W. (2016) Quorum
sensing systems differentially regulate the production of phenazine-1-carboxylic acid
in the rhizobacterium *Pseudomonas aeruginosa* PA1201. *Sci Rep*, 6,30352.

40. Price W.A., Dietrich L.E., Newman D.K. (2007) Pyocyanin alters redox
homeostasis and carbon flux through central metabolic pathways in *Pseudomonas*
*aeruginosa* PA14. *J Bacteriol*, 189, 6372-6381.

Dear reviewers :

Thank you for your letter and for the reviewers' comments concerning our manuscript entitled "A GntR family transcriptional repressor HutC sensing PCA regulated histidine catabolism in *Pseudomonas aeruginosa*" (ID: Spectrum 00824-25). These comments are all valuable and help us improve our manuscript. We have now performed a series of additional experiments and incorporated relevant data into the revised manuscript. Herein we have carefully addressed concerns and suggestions raised by the editors and reviewers and provide our point by point responses below.

Reviewers # 1 :

This is a potentially interesting paper by Yan and colleagues about the role of HutC in histidine catabolism. However, the paper is extremely difficult to read and follow, and I struggled to ascertain what exactly the authors were showing and how to interpret the data at several points. A non-exhaustive list of issues:

1、**Question:** The title is poor, requiring readers to know what a GntR family repressor is.

Response: Thank you for your good comments. A GntR family regulator is a member of the GntR bacterial transcription factor superfamily, characterized by a conserved N-terminal winged helix-turn-helix (wHTH) DNA-binding domain and a diverse C-terminal effector-binding/oligomerization domain. A transcriptional activator binds with the ligand and induces conformational changes of the transcriptional regulator which can bind to target promoters DNA normally, thereby activates gene expression. However, a transcriptional repressor binding with the ligand causes conformational changes of the transcriptional regulator which cannot bind to target promoters DNA, thereby inhibits gene expression. In our study, PCA ligand binds with the HutC effector protein and induces conformational changes of HutC protein structure. Target genes DNA dissociate with HutC transcriptional factor, thereby represses downstream gene expression.

To facilitate better understanding, we changed the title to “A GntR family transcriptional regulator HutC sensing PCA regulated histidine catabolism in *Pseudomonas aeruginosa*”.

2. There are many instances throughout where the grammar is incorrect. There are also lines (220-252) where individual words are split across two lines, which is extremely difficult to read.

Response: Thanks for the good suggestion. We have reviewed the entire paper and made revisions. The details are as follows:

1、 The original title “A GntR family transcriptional repressor HutC sensing PCA regulated histidine catabolism in *Pseudomonas aeruginosa*” was revised to “ A GntR family transcriptional regulator HutC sensing PCA regulated histidine catabolism in *Pseudomonas aeruginosa*”

2、 Line 38: the word “highlight” changed from original version paper “highlighte”.

3、 Lines 83-84: the sentence “we proved that the absorption peak of standard PCA aligns with the culture supernatant of $\Delta hutC$ strain by full-wavelength scanning and HPLC.” changed from original version paper “experimentally also proved that the absorption peak of standard PCA were aligns with the culture supernatant of $\Delta hutC$ strain at 380 nm by full-wavelength scanning”.

4、 Line 95: Due to the addition of some growth experiments, the original paper “nitrogen sources (10 mM histidine or urocanate)” was revised to “nitrogen sources (10 mM histidine, urocanate, PCA and 1-HP)” .

5、 Line 153: The original paper “(Mortensen *et al.*, 2014)” was revised to “[16]” .The original reference format is inconsistent with other references.

6、 Lines 224-234: Due to the addition of high-performance liquid phase(HPLC) experiments, the methods was added in lines 224-234. Please review.

7、 Line 269: The original paper “such as TCA cycle” was revised to “including TCA cycle” .The expression of the word 'including' is more accurate.

8、 Line 290: The original paper “these genes roles of the *hut* operon” was revised to “the roles of the *hut* operon” .

9、 Line 295: The original paper “exhibit the identical growth patterns” was revised to “exhibited the same growth patterns ” .

10、 Line 300: The original paper “suggested” was revised to “suggest”.

11、 Line 311: The original paper “was” was revised to “were”.

12、 Line 321: The word “*hutIG*” was revised to “*hutIG* promoter”.

13、 Line 322-323: The original paper “HutC can efficiently bind only to the *hutIG* promoter region (Fig. 3C), but cannot to *hutH*, *hutU*, and *hutF*” was revised to “HutC could efficiently bind only to the *hutIG* promoter region (Fig. 3C), but could not to *hutH*, *hutU*, and *hutF*”.

14、 Line 333: The original paper “To figure out what the brown matter is” was revised to “To identify the composition of this brown substance”.

15、 Line 361-362: The original paper “we further determined what the brown substance of Δ *hutC* strain supernatant might be through full-wavelength scanning.” was revised to “we further determined the brown substance in the Δ *hutC* strain supernatant through full-wavelength scanning.”.

16、 Lines 367-371: Due to the addition of high-performance liquid phase(HPLC) experiments, the results were added in lines 367-371. Please review.

17、 Lines 372-373: The original paper “So we speculated that PCA may be essential for the transcriptional regulation mediated by HutC” was revised to “Based on previous studies, we speculated that PCA may be a potential effector of HutC.”.

18、 Lines 376-387: In order to more fully assess the impact of PCA as an effector on HutC-mediated regulation of gene expression, we have added the experimental results of luminescence screening assays and growth assay in 376-387. Please review.

3. Figure 1A - the choice of red and green would make the volcano plot uninterpretable for someone with red/green colorblindness

Response: Thanks for the good suggestion. We have changed the color of the volcano plot to red/blue (Fig. 1A).

4. Figure 2 - the symbols are too small, as is the text, and it is difficult to figure out which lines are which strains. This would be a good location for color!

Response: Thanks for the good suggestion. Indeed, the symbols and the text are too small to see clearly. We have changed the color of the symbols and enlarged the text (Fig. 2).

5. Figure 3 - panel A does not show promoter activities. (In fact it is duplicated from Figure 2)

Response: We are very sorry to place the wrong figure of 3A. We have placed the correct figure of *hutG* promoter activities (figure 3A).

6. Figure 3C - it seems like there is a gel shift with no added protein? Also, which is the tested *hutU* promoter so much longer than the others?

Response: Indeed, it seems like there is a gel shift with no added protein. We have revised the image of figure 3C. The *hutU* promoter is 324 bp. But the *hutH*, *hutF* and *hutG* promoter is merely 95 bp, 99bp and 116bp respectively. So the molecular weight of *hutU* promoter is much larger than the others' promoter, and thus the *hutU* migrate more slowly and appear as an upper band under same electrophoresis conditions.

7. Figure 4A - the legend talks about three strains but only two are shown.

Response: We are very sorry to make such a basic mistake. we found that knocking out *hutC* exhibited a dramatic accumulation of dark brown substance compared with the WT PAO1 in LB medium. So we only conducted comparative experiments of two strains. We have modified the legend of figure 4A and deleted the $\Delta hutC/p-hutC$ strain.

8. Figure 4B legend. What is "Pancreatic metabolic profile"???

Response: We are very sorry to express mistake. "Pancreatic metabolic profile" should be expressed as "The metabolic profile".

9. Figure 4C - the text is way too small.

Response: Thanks for the good suggestion. We have enlarged the text of

figure 4C.

10. Figure 5. The text is appropriately circumspect about the potential role of PCA but the title of the figure is overly definitive. It's possible that PCA is one contributor to HutC regulation.

Response: Thanks for the good comments and provided us with the direction for next research. In our study, 167 differential metabolites were screened between the WT PAO1 and $\Delta hutC$ mutants using non-targeted metabolomics. However, combined with phenotypic assays, full-wavelength scanning, HPLC, EMSA, promoter activity assay and growth assay, these results indicate PCA is relatively effective to HutC regulation compared to other substances. So PCA is one contributor to HutC regulation. Next, we will study the effects of other potential substances on HutC-mediated genes expression.

Reviewer #2 (Comments for the Author):

The work by Cui and colleagues, presents a study on the transcriptional regulation of the hut operon by HutC, a GntR family transcriptional repressor, in *Pseudomonas aeruginosa*. They apply Transcriptomics, gene deletion and complementation, EMSA, metabolomics and functional assays to elucidate the regulatory mechanisms of hutC in *Pseudomonas*. They show the role of hutC in histidine catabolism and its interaction with phenazine-1-carboxylic acid (PCA) as a regulatory ligand

Comments and Suggestions.

1、 **Question:** There are a few errors in grammar, sentence structures and word choice which can be corrected to improve clarity.

Response: Thanks for the good suggestion. We have carefully reviewed the paper and made modifications to grammar, sentence structures, and word choices. The details are as follows:

1、 The original title “A GntR family transcriptional repressor HutC sensing PCA regulated histidine catabolism in *Pseudomonas aeruginosa*” was revised to “A GntR family transcriptional regulator HutC sensing PCA regulated histidine catabolism in *Pseudomonas aeruginosa*”

2、 Line 38: the word “highlight” changed from original version paper “highlighte”.

3、 Lines 83-84: the sentence “we proved that the absorption peak of standard PCA aligns with the culture supernatant of $\Delta hutC$ strain by full-wavelength scanning and HPLC.” changed from original version paper “experimentally also proved that the absorption peak of standard PCA were aligns with the culture supernatant of $\Delta hutC$ strain at 380 nm by full-wavelength scanning”.

4、 Line 95: Due to the addition of some growth experiments, the original paper “nitrogen sources (10 mM histidine or urocanate)” was revised to “nitrogen sources (10 mM histidine, urocanate, PCA and 1-HP)” .

- 5、 Line 153: The original paper “(Mortensen *et al.*, 2014)” was revised to “[16]” .The original reference format is inconsistent with other references.
- 6、 Lines 224-234: Due to the addition of high-performance liquid phase(HPLC) experiments, the methods was added in lines 224-234. Please review.
- 7、 Line 269: The original paper “such as TCA cycle” was revised to “including TCA cycle” .The expression of the word 'including' is more accurate.
- 8、 Line 290: The original paper “these genes roles of the *hut* operon” was revised to “the roles of the *hut* operon” .
- 9、 Line 295: The original paper “exhibit the identical growth patterns” was revised to “exhibited the same growth patterns ” .
- 10、 Line 300: The original paper “suggested” was revised to“suggest”.
- 11、 Line 311: The original paper “was” was revised to“were”.
- 12、 Line 321: The word “*hutIG*” was revised to“*hutIG* promoter”.
- 13、 Line 322-323: The original paper “HutC can efficiently bind only to the *hutIG* promoter region (Fig. 3C), but cannot to *hutH*, *hutU*, and *hutF*” was revised to“HutC could efficiently bind only to the *hutIG* promoter region (Fig. 3C), but could not to *hutH*, *hutU*, and *hutF*”.
- 14、 Line 333: The original paper “To figure out what the brown matter is” was revised to“To identify the composition of this brown substance”.
- 15、 Line 361-362: The original paper “we further determined what the brown substance of Δ *hutC* strain supernatant might be through full-wavelength scanning.” was revised to“we further determined the brown substance in the Δ *hutC* strain supernatant through full-wavelength scanning.”.
- 16、 Lines 367-371: Due to the addition of high-performance liquid phase(HPLC) experiments, the results were added in lines 367-371. Please review.
- 17、 Lines 372-373: The original paper “So we speculated that PCA may be essential for the transcriptional regulation mediated by HutC” was revised to “Based on previous studies, we speculated that PCA may be a potential effector of HutC.”.

18、 Lines 376-387: In order to more fully assess the impact of PCA as an effector on HutC-mediated regulation of gene expression, we have added the experimental results of luminescence screening assays and growth assay in 376-387. Please review.

2、 Question: From the transcriptomic, what was the expression of the urocanate transporter, and how does it explain the growth patterns of the Wild type and the hut mutants in urocanate?

Response: Thanks. From the transcriptomic, the expression of the urocanate transporter was downregulated by transcription factors HutC. Urocanate is an intermediate product of the hut pathway, and its accumulation cannot provide downstream carbon shelves and energy. Cells are unable to obtain sufficient carbon sources and energy to sustain growth. when urocanate was used as the sole source of carbon/nitrogen source, urocanate transporters may ingest urocanate into the cell. However, the lack of HutU(the first intermediate of the histidine degradation pathway) in the hut mutants means that urocanate cannot be metabolized. Urocanate accumulates rapidly inside the cell and causes toxicity accumulation, pH imbalance, metabolic obstruction. Finally, the hut mutants cannot (or can only grow extremely weakly) grow with urocanate as the sole carbon/nitrogen source. In contrast, the *hut* genes exist in the wild type. The *hutU* gene(urocanate hydratase) rapidly converts urocanate into imidazolone propionate(IP), and ring cleavage of IP to yield formiminoglutamate (FIG) by *hutI*. The imino group of FIG is first hydrolyzed to yield ammonia and formylglutamate (FG) by *hutF*. FG is then hydrolyzed to give formate and glutamate by *hutG*. Finally the wild type enters downstream metabolic pathways and ultimately produces glutamate (nitrogen source) and formic acid, ammonia, ATP (carbon source and energy).

3、 Question: What negative control was used for the EMSA binding assay for assessing PCA disruption of HutC binding?

Response: Thanks for the good suggestion. We used histidine as negative control for assessing PCA disruption of HutC binding(Fig. 5C).

4、 **Question:**Did exogenous addition of PCA and 1-HP affect the production of the brown pigment in the *hutC* mutant or the WT? Did you observe any phenotypic characteristics in the wild type or *hut* mutants, such as changes in growth pattern, supporting the role of PCA and 1-HP in *hutC* pathway?

Response: We sincerely thank the reviewer for the good suggestions. We have measured the growth pattern in M9 medium supplemented with 5 mM PCA and 1-HP among the wild type, *hutC* mutants and their complemented strains. Our growth showed that exogenous addition of PCA and 1-HP does not affect the production of the brown pigment in the *hutC* mutant and the WT. The color of culture medium is same with and without the addition of PCA and 1-HP. The production of the brown pigment is caused by *hutC* gene knockout. However, the growth of the Δ *hutC* mutants have significant growth defects compared to the wild type when 10 mM PCA were added (Fig. 5D). Meanwhile, this growth defect was restored to the WT levels by introducing the corresponding complemented plasmid to these mutants. However, the Δ *hutC* mutants grew like the WT when 10 mM 1-HP were added(Fig. S5). These results indicate that PCA as a HutC regulatory ligand binding to HutC transcription factor participates in the regulation of Hut pathway.

5、 **Question:**While it is stated that PCA disrupts HutC binding, the biochemical mechanism is speculative and not explored in detail, as there is less information on the structural binding site or conformation changes. Please provide more details on PCA production and its environmental triggers or conditions.

Response: Thanks for the good suggestion. We have already monitored the activity of *hutG-lux* in the Δ *hutC* mutant strain in the presence or absence of PCA and compared to WT PAO1. We observed that in WT PAO1 and the complemented strain, PCA strongly induced the activity of *hutG* gene. No significant difference in *hutG* expression could be found in the Δ *hutC* mutant strain (Fig. 5C). The result also suggest that PCA promote the HutC-mediated regulation of gene expression.

Re: Spectrum00816-25R1 (**A GntR family transcriptional repressor HutC sensing PCA regulated histidine catabolism in *Pseudomonas aeruginosa***)

Dear Dr. cui jia:

Thank you for the privilege of reviewing your work. Below you will find my comments, instructions from the Spectrum editorial office, and the reviewer comments.

Dear Jia,

Based on the reviewers comments the manuscript needs still work. Please carefully address all the suggestions and comments provided, especially by reviewers 1 and 3, and resubmit again.

Revision Guidelines

Sincerely,
Monica Cartelle Gestal
Editor
Microbiology Spectrum

Reviewer #1 (Comments for the Author):

The authors have largely addressed the material issues that I had with the initial manuscript, although the EMSA figures still are

not very compelling, particularly for figure 5 -- it looks like there is less of everything in the 2mM PCA lane. In addition, there is no label for Fig 5B and the concentration units for PCA is not stated in the panel.

There are still issues of writing throughout. In the interest of being helpful, here are some specific suggestions:

1. A better title might be "The GntR family regulator HutC senses PCA to regulate histidine catabolism in *Pseudomonas aeruginosa*"
2. Line 23-24, remove "by untargeted metabolomics analysis and EMSA"
3. Lines 71-72. Please clarify what is more difficult?
4. RNA-seq method -- how many biological replicates? Why an OD of 0.6?
5. line 282. Why is this remarkable?
6. line 284, should be "limitation to growth"; line 288, should be "homology analysis"
7. Line 372, what previous studies? (citations please)

Reviewer #2 (Comments for the Author):

All comments and suggestions have been addressed

Reviewer #3 (Comments for the Author):

Thank you for the effort put into the review and for answering the various questions and queries suggested. This manuscript could benefit from further corrections and might need another set of eyes to peruse and note the several grammatical and other sentence construction errors. A non-exhaustive list is included below:

1. Page 1, line 24. Spell out EMSA when it is first used (Abstract)
2. Page 3, lines 62, 65, 71-72; page 4, line 117; page 5, lines 132, 134, etc. Check sentence construction, punctuation, capitalization, subject-verb agreement, etc.
3. Page 4, line 114. Genus and species in italics
4. Page 5, line 124. Unsure what 'LB agar substrates' means.
5. Page 5, line 136. Unsure what 'processed in twice' means
6. Page 7, line 180. Change elctrotransformed to electroporated
7. Page 7, line 184. Include source of 96-well plates and state whether round- or flat-bottomed
8. Page 8, Phenazine Analysis: check tense. Also, lines 236 and 248. Remember that methods are written in the past tense.
9. For Results section, be careful about how much of the literature is cited before narrating the results from the experiments, e.g., page 10, lines 279-287. Also Results must be recorded in the past tense.
10. Page 11, line 314. It would be useful to note the p-value here.
11. Pages 12-13, lines 352-363. Please check the sentence construction in this section of the Results.
12. Page 13, line 386. Change 'Is' to 'is'
13. Page 14 (Discussion). Please check sentence construction, especially lines 389, 393, 395, 404, 407, 408 and 414.

Dear reviewers :

Thank you for your letter and for the reviewers' comments concerning our manuscript entitled "A GntR family regulator HutC senses PCA to regulate histidine catabolism in *Pseudomonas aeruginosa*" (ID: Spectrum 00816-25R1). These comments are all valuable and help us improve our manuscript. Herein we have carefully addressed concerns and suggestions raised by the editors and reviewers and provide our point by point responses below.

Reviewers # 1 (Comments for the Author):

1. **Question:** there is no label for Fig 5B and the concentration units for PCA is not stated in the panel.

Response: Thank you for your good comments. We are very sorry to express mistake. We have added the label for Fig 5B and the concentration units for PCA. Please refer to the uploaded Fig 5 for details.

2. **Question:** A better title might be "The GntR family regulator HutC senses PCA to regulate histidine catabolism in *Pseudomonas aeruginosa*"

Response: Thank you for your good suggestion. We fully accept your suggestion. The original title "A GntR family transcriptional repressor HutC sensing PCA regulated histidine catabolism in *Pseudomonas aeruginosa*" has been revised to the title "The GntR family regulator HutC senses PCA to regulate histidine catabolism in *Pseudomonas aeruginosa*".

3. **Question:**Line 23-24, remove "by untargeted metabolomics analysis and EMSA"

Response: Thanks for the good suggestion. We have removed "by untargeted metabolomics analysis and EMSA" in line 23-24.

4. **Question:**Lines 71-72. Please clarify what is more difficult?

Response: Thanks. The Hut (Histidine Utilization) operon allows bacteria like *Klebsiella pneumoniae* and *Bacillus subtilis* to use the amino acid histidine as a source of carbon, nitrogen, and energy. Its importance as a regulatory paradigm comes from its sophisticated control system. It serves as a natural "reporting system" that can be utilized to intuitively investigate how cells

integrate dual signals from carbon (energy) and nitrogen (nutrient) sources to make optimal gene expression decisions. *E. coli*, the model organism for so many studies, lacks the hut operons. Although *E. coli* possesses other dually regulated systems, such as the lac operon, which is modulated by both CRP and lactose induction, the lac system primarily reflects carbon source status and does not involve nitrogen regulation. When studying carbon-nitrogen co-regulation, researchers are more difficult in *E. coli* and compelled to resort to other, more complex or less intuitive systems.

5. **Question:**RNA-seq method -- how many biological replicates? Why an OD of 0.6?

Response: Thanks . Three biological replicates of each sample are performed in the RNA-seq method to ensure the reliability, accuracy, and statistical significance of the research findings. RNA-seq technology can comprehensively analyze bacterial transcriptome and reveal the differential gene expression of genes in specific physiological states. An OD of 0.6 was selected as the harvesting point primarily to ensure that the bacteria were in the logarithmic growth phase, which is the standard practice for transcriptomic analysis for many bacteria. In this study, an OD of 0.6 also corresponds to the logarithmic phase for *Pseudomonas aeruginosa*. During logarithmic growth phase, metabolically active, and their gene expression profile of bacterial cells is stable reproducible, and avoiding stationary phase effect.

6. **Question:** line 282. Why is this remarkable?

Response:Thanks for the good question. Urocanate retains nearly all the carbon and nitrogen source derived from its precursor. In theory, *Pseudomonas* can grow normally in a medium with urocanate as the sole substrate. However, due to the inherent inability to directly assimilate exogenous urocanate, *P. putida* encounters a significant limited grow with urocanate as the sole substrate. However, *P. aeruginosa* harbors the putative urocanate transporter gene, the exogenous urocanate can be assimilated, absorbed and growth normally. The appearance of remarkably mainly reminds people to pay attention to the evolutionary distinction between *Pseudomonas aeruginosa* and other bacteria.

7. Question: line 284, should be "limitation to growth"; line 288, should be "homology analysis"

Response: Thanks for the good suggestion. We have revised the original paper "limited to growth" to "limitation to growth" in line 284. The original paper "Based on homologous analysis " was revised to "Based on homology analysis" in line 288.

8. Question: Line 372, what previous studies? (citations please)

Response: Thanks for the good suggestion. We have added the references 17.

Reviewer #2 (Comments for the Author):

All comments and suggestions have been addressed

Reviewer #3 (Comments for the Author):

Thank you for the effort put into the review and for answering the various questions and queries suggested. This manuscript could benefit from further corrections and

might need another set of eyes to peruse and note the several grammatical and other sentence construction errors. A non-exhaustive list is included below:

1. **Question:** Page 1. line 24. Spell out EMSA when it is first used (Abstract)

Response: Thanks for the good suggestion. We have revised “EMSA” to “Electrophoretic mobility shift assay” in line 21.

2. **Question:** Page 3, lines 62, 65, 71-72; page 4, line 117; page 5, lines 132, 134, etc. Check sentence construction, punctuation, capitalization, subject-verb agreement, etc.

Response: Thanks for the good suggestion. The details are as follows:

1、 **Line 62:** The original paper “ And ring cleavage of IP to yield formimino-glutamate (FIG) by HutI.” was revised to “And ring of IP is cleaved to yield formimino-glutamate (FIG) by HutI.”.

2、 **Line 65:** The original paper “whereas in enteric bacteria such as *Salmonella*, *Klebsiella* and gram-positive bacteria,” was revised to “whereas in enteric bacteria (such as *Salmonella*, *Klebsiella*) and gram-positive bacteria,” .

3、 **Line 71-72:** The original paper “futile production or excessive catabolism are unwarranted” was revised to “futile production or excessive catabolism is unwarranted” .

4、 **Lines 117:** The original paper “The upstream primer pairs of pEX-*hutC*-up-F/R and The downstream primer pairs” was revised to “The upstream primer pairs of pEX-*hutC*-up-F/R and the downstream primer pairs” .

5、 **Line 132:** The original paper “using the RiboCop rRNA Depletion Kit for Mixed Bacterial Samples” was revised to “using the RiboCop rRNA Depletion Kit for mixed bacterial samples” .

6、 **Line 134:** The original paper “using of total RNA” was revised to “using total RNA ” .

3. **Question:** Page 4, line 114. Genus and species in italics

Response: Thanks for the good suggestion. We have revised “*P. aeruginosa*” to “*P. aeruginosa*” in line 114.

4. **Question:** Page 5, line 124. Unsure what 'LB agar substrates' means.

Response: Thanks. LB is generally interpreted as Luria-Bertani medium; however, according to its inventor Giuseppe Bertani, the name is derived from the English term "lysogeny broth". LB agar substrates has become the most common and widely used medium for bacterial amplification in microbiology and molecular biology experiments due to its advantages of simple composition and easy preparation. It typically contains three key ingredients:tryptone, yeast extract and NaCl.

5. **Question:** Page 5, line 136. Unsure what 'processed in twice' means

Response: Thanks. "Each sample was processed in twice" means that the same original sample independently repeats the library construction process in twice. The purpose is to control technical errors and improve data reliability.

6. **Question:** Page 7, line 180. Change elctrotransformed to electroporated

Response: Thanks for the good suggestion. We have revised "elctrotransformed " to "electroporated" in line 180.

7. **Question:** Page 7, line 184. Include source of 96-well plates and state whether round- or flat-bottomed

Response: Thanks for the good suggestion. We have revised "transferred to clear-bottom 96-well plates " to "transferred to 96-well round-bottom plates with clear-bottoms" in line 184. The statement is more suitable for scientific writing and standardization.

8. **Question:** Page 8, Phenazine Analysis: check tense. Also, lines 236 and 248. Remember that methods are written in the past tense.

Response: Thanks for the good suggestion. We have carefully checked the paragraph and made modifications in line 217-223, 236 and 248. We changed the tense to the past tense. Please refer to the marked-up file for details.

9. **Question:** For Results section, be careful about how much of the literature is cited before narrating the results from the experiments, e.g., page 10, lines 279-287. Also Results must be recorded in the past tense.

Response: Thanks for the good suggestion. We have carefully checked the number of references cited throughout the full text. The literatures are cited

from 17-21 in line 257-258 of the results section. So the literatures are cited from 22 -23 in lines 279-287. We also changed the present tenses to the past tenses in 267 and 269.

10. **Question:** Page 11, line 314. It would be useful to note the p-value here.

Response: Thanks for the good suggestion. We have performed grayscale value analysis of the protein bands using Image J and added the p-value ($p < 0.01$) in line 314.

11. **Question:** Pages 12-13, lines 352-363. Please check the sentence construction in this section of the Results.

Response: Thanks for the good suggestion. We have carefully checked this section and discovered some errors in line 352-363. The details are as follows:

1、 **Line 352:** The original paper “ HutC, as a typical the GntR family regulator ” was revised to “ HutC, as a typical GntR family regulator”.

2、 **Line 354:** The original paper “And that regulation of gene expression is dependent on the binding of an effector molecule ” was revised to “Its regulation of gene expression depends on the binding of an effector molecule ”.

3、 **Line 358-359:** The original paper “ phenazine compounds impart different pigments to *P. aeruginosa* cultures and the $\Delta hutC$ mutants produced a large amount of dark brown substance ” was revised to “ phenazine compounds imparted different pigments to *P. aeruginosa* cultures and the $\Delta hutC$ mutants produced a large amount of dark brown substance”.

4、 **Line 360:** The original paper “ phenazine compounds were also screened out ” was revised to “ phenazine compounds were also identified ”. The word 'identified' expresses more accurately and professionally than 'screened out'.

12. **Question:** Page 13, line 386. Change 'Is' to 'is'

Response: Thanks for the good suggestion. We have revised “Is” to “is” in line 386.

13. **Question:** Page 14 (Discussion). Please check sentence construction, especially lines 389, 393, 395, 404, 407, 408 and 414.

Response: Thanks for the good suggestion. We have carefully checked this section and discovered some errors. The details are as follows:

1、Line 389-390: The original paper “ The ability of some opportunistic pathogens can adapt smoothly to various environments is most likely a main cause of the associated disease ” was revised to “The ability of some opportunistic pathogens to adapt smoothly to various environments is most likely a main cause of the associated diseases ”.

2、Line 389-390: The original paper “ The ability of some opportunistic pathogens can adapt smoothly to various environments is most likely a main cause of the associated disease ” was revised to “The ability of some opportunistic pathogens to adapt smoothly to various environments is most likely a main cause of the associated diseases ”.

3、Line 390-391: The original paper “ Amino acids as nutrient sources are important for many bacterial proliferation and metabolic processes ” was revised to “Amino acids, as nutrient sources, are important for the proliferation and metabolic processes of many bacteria ”.

4、Line 393: The original paper “and thus both the import/uptake and its degradation/secretion” was revised to “and thus both the import/uptake and degradation/secretion ”.

5、Line 395-396: The original paper “we coupled reverse genetics with bioinformatics to identify a *hut* operon participated in histidine catabolism in *P. aeruginosa* ” was revised to “we coupled reverse genetics with bioinformatics to identify a *hut* operon that participates in histidine catabolism in *P. aeruginosa*”.

6、Line 404: The original paper “our data showed that ” was revised to “our data identified that”.

7、Line 407: The original paper “, especially the $\Delta hutH$ mutant grow relatively more slowly ” was revised to “; especially, the $\Delta hutH$ mutant grew relatively more slowly ”.

8、 Line 408: The original paper “histidine catabolismis” was revised to “histidine catabolism”.

9、 Line 412-415: The original paper “*P. aeruginosa* and *Klebsiella pneumoniae* harboring a urocanate transporter gene are thus able to utilize urocanate [32]. Our findings also indicated that the deletion of *hutCUHIGF* unable to efficiently utilize urocanate for growth” was revised to “*P. aeruginosa* and *Klebsiella pneumoniae* harboring a urocanate transporter gene were thus able to utilize urocanate [32]. Our findings also indicated that the deletion of *hutCUHIGF* were unable to efficiently utilize urocanate for growth”.

10、 Line 438: The original paper “The addition of 2 mM PCA can directly prevent the binding of HutC to *hutIG* promoter DNA” was revised to “The addition of 2 mM PCA could directly prevent the binding of HutC to *hutIG* promoter DNA”.

Re: Spectrum00816-25R2 (**A GntR family regulator HutC senses PCA to regulate histidine catabolism in *Pseudomonas aeruginosa***)

Dear Dr. cui guoyan:

Your manuscript has been accepted, and I am forwarding it to the ASM production staff for publication. Your paper will first be checked to make sure all elements meet the technical requirements. ASM staff will contact you if anything needs to be revised before copyediting and production can begin. Otherwise, you will be notified when your proofs are ready to be viewed.

Sincerely,
Monica Cartelle Gestal
Editor
Microbiology Spectrum

Reviewer #1 (Comments for the Author):

The authors have responded to my comments. The title would still be better if it started with "The" rather than "A"